# Examining the Human Perceptibility of
# Black-Box Adversarial Attacks on Face Recognition

**Benjamin Spetter-Goldstein** [1]  **Nataniel Ruiz** [1]  **Sarah Adel Bargal** [1]

## Abstract

The modern open internet contains billions of public images of human faces across the web, especially on social media websites used by half the world's population. In this context, Face Recognition (FR) systems have the potential to match faces to specific names and identities, creating glaring privacy concerns. Adversarial attacks are a promising way to grant users privacy from FR systems by disrupting their capability to recognize faces. Yet, such attacks can be perceptible to human observers, especially under the more challenging black-box threat model. In the literature, the justification for the imperceptibility of such attacks hinges on bounding metrics such as $\ell_p$ norms. However, there is not much research on how these norms match up with human perception. Through examining and measuring both the effectiveness of recent black-box attacks in the face recognition setting and their corresponding human perceptibility through survey data, we demonstrate the trade-offs in perceptibility that occur as attacks become more aggressive. We also show how the $\ell_2$ norm and other metrics do not correlate with human perceptibility in a linear fashion, thus making these norms suboptimal at measuring adversarial attack perceptibility.

## 1. Introduction

Face recognition (FR) technology is becoming increasingly common and presents many chances to be abused. Biometric FR systems are now regularly used in smartphones, social media apps, and surveillance systems by millions of people (Owen, 2021). Recent widespread use of social media websites has lead to billions of public images of human faces available for anyone to access online. Many of these faces are also linked to other identifiable information, such as names, birthdays, email addresses (Facebook, 2021). This presents ample opportunity for abuse, with many prior large-scale examples such as FindFace, an FR system designed for the Russian social media website VK where users could upload a photo of a face and link it back to the corresponding profile. Police used the website to identify suspects and protesters to crack down on political dissidents, and malicious web users were able to find and dox (leak sensitive personal information of) anonymous internet celebrities (Guarino, 2016).

Recent successful investigations have gone into the topics of using adversarial attacks in order to disrupt FR systems. These attacks are adapted from the ImageNet classification task and propose modifications for the FR setting as well as metrics for evaluating the strength and quality of these attacks (Dong et al., 2019; Goswami et al., 2018; Shan et al., 2020; Yang et al., 2020).

The need for such attacks to be imperceptible is twofold and especially important in this scenario. First, if the attack is not imperceptible, an FR system could potentially detect that the image had been modified and alert the owner of the system. Second, a user that attacks one of their own images and uploads it onto the web would like the image to conserve its original perceptual quality which can be greatly impacted by an adversarial attack.

Most attacks in the literature justify their imperceptibility by bounding the $\ell_p$ norm of the attack or reporting image quality metrics such as Structural Dissimilarity (DSSIM) (Shan et al., 2020). Some attacks use the full $\ell_p$ norm budget for all attacks (Andriushchenko et al., 2019), whereas others terminate when the attack is successful, thus using less than the full budget for some attacks (Guo et al., 2019; Ilyas et al., 2018a;b). We hypothesize that different attacks have a different human perceptibility profile depending on these and other variables. In this work, we have examined recent state-of-the-art adversarial attacks, adapted them to attack FR systems, gathered metrics on their performance, and tested their perceptibility to human observers. The contributions of our work are as follows:

---

[1]Department of Computer Science, Boston University, Boston, Massachusetts, United States. Correspondence to: Benjamin Spetter-Goldstein <benjisg@bu.edu>.

*Accepted by the ICML 2021 workshop on A Blessing in Disguise: The Prospects and Perils of Adversarial Machine Learning.* Copyright 2021 by the author(s).

1. The attacks we present have originally been tested on an ImageNet classification task. We test them in the face verification scenario and an in-the-wild face dataset, thus measuring how well they adapt to a different data distribution.

2. We demonstrate trade-offs between attack success rate, magnitude, and human perceptibility. In particular, no attack achieves a 100% success rate while escaping human detection.

3. Ideally, attack magnitude metrics should have a linear relationship with human perceptibility. We observe that the $\ell_2$ attack metric and the DSSIM image quality metrics are not linearly correlated to human perceptibility and thus are sub-optimal at measuring attack perceptibility. We encourage future work on attacks bounded by other metrics and work that explores metrics that are more correlated with human perception.

## 2. Background and Related Work

Previous work has examined the limitations of $\ell_p$ norms as well as the Structural Similarity (SSIM) metric using surveys of human perceptibility. Some works focus on attacks operating on the ImageNet classification task using the white-box threat model(Sen et al., 2019; Sharif et al., 2018), while others examine black-box attacks as well and propose new computational metrics meant to act more in line with human perception (Laidlaw et al., 2020; Quan, 2020). Our work focuses on the black-box threat model in the context of the face verification task and seeks specifically to determine if attacks can achieve perfect success in a query-limited setting and still remain imperceptible, as well as how these leading attacks' tradeoffs in perceptibility and success compare against one another. We additionally continue the examination of $\ell_p$ norms by examining how well the $\ell_2$ norm performs in this setting, as well as whether DSSIM performs significantly different from the $\ell_2$ norm.

## 3. Method

In this section, we present the methods and parameters considered when creating adversarial examples for FR systems.

**Face Verification Adversarial Examples.** An adversarial example is any input to a machine learning model that an attacker has deliberately modified in order to cause the model to produce a false output. Representing the deep learning model as $\mathcal{F}$, we seek to add some perturbation $\eta$ to a given image $x$ such that $\mathcal{F}(x) \neq \mathcal{F}(x + \eta)$. In the case of image classifiers, this is typically performed by causing the classifier to assign a false label with changes that decrease confidence in the true label while increasing confidence in one or more false labels. When working with an FR

---

**Algorithm 1** Face Recognition NES (FR-NES)

**Input:** The attack objective function $\mathcal{F}(\mathbf{x}^*)$; the original face images $x_{init}$, $y_{init}$; the maximum allowed queries $T$; the decision boundary $d_b$; the learning rate $\eta$; the total perturbation bound $\epsilon$

$x_0, y \leftarrow x_{init}, y_{init}$
$d_0 \leftarrow \ell_2(\mathcal{F}(x_0), \mathcal{F}(y))$
**for** $t = 1 \ldots T$ **do**
    $x_t \leftarrow x_{t-1} + \eta \cdot \epsilon$
    $d_t \leftarrow \ell_2(\mathcal{F}(x_t), \mathcal{F}(y))$
    **if** $d_t \geq d_b$ **then**
        **break**
    **end if**
**end for**

---

system, we reduce the space of our outputs to two labels, $l \in \{0, 1\}$—either two faces are classified as a match, indicating they belong to the same person, or they are classified as not a match, indicating they belong to different people. We attack a pair of faces, designating one image of the pair as our source image, which we will use to see how successful the attack is. We designate the other image in the pair as our target image, which we will slowly modify until it produces a different output from the FR system.

Since the FR network produces a set of features as its output, we compare these features using our chosen distance metric, in this case the $\ell_2$ norm, and halt our attack once this distance rises above a pre-selected threshold $d_b$. Thus, in the FR setting, given a pair of matching face images $(x, y)$ where $\ell_2(\mathcal{F}(x), \mathcal{F}(y)) < d_b$, we seek to produce $x + \eta$ such that $\ell_2(\mathcal{F}(x + \eta), \mathcal{F}(y)) \geq d_b$. The parameter that controls the maximum norm of the perturbation of the attack for all non-SimBA attacks is referred to as $\epsilon$. To avoid exceeding $\epsilon$, we specify our step size $\eta$ such that at each iteration we take one step of size $\eta \cdot \epsilon$. Algorithm 1 presents NES adapted to the FR setting in this manner. The calculated perturbation of the final attack, known as the magnitude of the attack, is taken using the $\ell_2$ norm of the difference between the adversarial attack and the original unaltered target image. Since many attacks finish within the query limit, this is often different from $\epsilon$.

The Human Accuracy of a given attack is calculated by evaluating the rate at which a majority of human observers are able to correctly identify an example as digitally altered or not altered after being shown an example in which the given attack is very perceptible. Upon assessing the perceptibility of all attacks, we seek to examine to what extent there exists a tradeoff between the success rate and imperceptibility of each attack, whether any attack manages to achieve 100% while remaining imperceptible, how well the $\ell_2$ magnitude and DSSIM correlate with human perception, and whether any attack demonstrates superiority in all metrics.

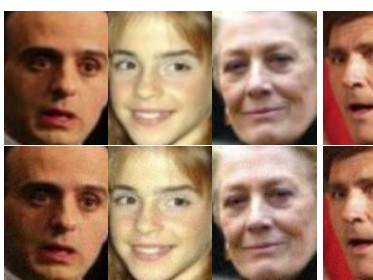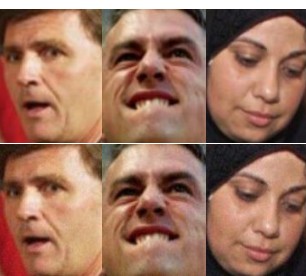

*Figure 1.* Most (left) and least (right) often successfully identified faces. Unaltered examples are in the top row with adversarial examples on the bottom row (produced using NES, $\epsilon = 18$).

*Table 1.* Statistics across attacks and $\epsilon$ values

| | $\epsilon$ | SUCCESS RATE | HUMAN ACCURACY | AVERAGE MAGNITUDE | AVERAGE DSSIM | AVERAGE QUERIES |
|---|---|---|---|---|---|---|
| **NES** | 10.0 | 0.760 | 0.47 | 4.4945 | 0.013 | 5286.62 |
| | 12.0 | 0.852 | 0.47 | 4.7149 | 0.015 | 4331.32 |
| | 14.0 | 0.916 | 0.56 | 5.37 | 0.018 | 3629.4 |
| | 16.0 | 0.956 | 0.64 | 5.37 | 0.02 | 3069.3 |
| | 18.0 | 0.986 | 0.61 | 5.6595 | 0.023 | 2643.98 |
| | 20.0 | 0.998 | 0.6 | 5.9206 | 0.025 | 2302.18 |
| | 25.0 | 1 | 0.68 | 6.568 | 0.031 | 1764.58 |
| | 30.0 | 1 | 0.66 | 7.1447 | 0.038 | 1447.58 |
| | 50.0 | 1 | 0.82 | 9.1433 | 0.062 | 905.06 |
| **Bandits** | 12.0 | 0.892 | 0.42 | 4.9648 | 0.016 | 3796.17 |
| | 14.0 | 0.948 | 0.49 | 5.3463 | 0.02 | 3028.93 |
| | 16.0 | 0.982 | 0.6 | 5.8055 | 0.024 | 2435.49 |
| | 18.0 | 0.998 | 0.56 | 6.2755 | 0.028 | 1996.9 |
| | 20.0 | 0.998 | 0.54 | 6.7456 | 0.033 | 1674.36 |
| **Square** | 12.0 | 0.858 | 0.71 | 8.4917 | 0.033 | 3759.85 |
| | 14.0 | 0.93 | 0.6 | 10.0201 | 0.047 | 2790.76 |
| | 16.0 | 0.96 | 0.95 | 11.6048 | 0.052 | 2084.84 |
| | 18.0 | 0.99 | 0.99 | 13.2445 | 0.056 | 1530.57 |
| | 20.0 | 0.998 | 0.97 | 14.9723 | 0.059 | 1120.89 |
| **SimBA** | 12.0 | 0.406 | 0.56 | 3.5122 | 0.01 | 2977.72 |
| | 14.0 | 0.634 | 0.55 | 4.0871 | 0.012 | 3596.51 |
| | 16.0 | 0.8 | 0.55 | 4.3857 | 0.014 | 3977.15 |
| | 18.0 | 0.918 | 0.55 | 4.6292 | 0.015 | 4201.26 |
| | 20.0 | 0.978 | 0.59 | 4.7765 | 0.016 | 4285.86 |

## 4. Experiments

In this section, we present the experiments performed to evaluate the effectiveness, efficiency, and human perceptibility for each attack method. We study our adaptations of the NES (Ilyas et al., 2018a), Bandits-TD (Ilyas et al., 2018b), SimBA (Guo et al., 2019), and Square (Andriushchenko et al., 2019) attacks for face recognition.

### 4.1. Adversarial Attack Experimental Setup

We use a random subset of 500 pairs of faces from the Labeled Faces in the Wild (LFW) dataset (Huang et al., 2007) for testing. The targeted model is InceptionRes-netV1(Schroff et al., 2015) trained on VGGFace2(Cao et al., 2017). The faces were detected and cropped from LFW using the MTCNN network (Zhang et al., 2016). The threshold $d_b = 1.14$ for verifying a face pair was selected using Precision-Recall analysis over the dataset. We use the default ground truth parameters for all attacks. We set the query limit to 10,000, except for the SimBA attack, in which the query limit controls the maximum perturbation of the attack and is adjusted accordingly. The $\epsilon$ of each attack was selected from $\epsilon = \{12, 14, 16, 18, 20\}$. For NES we perform a more extensive evaluation and include $\epsilon = \{10, 25, 30, 50\}$.

### 4.2. Human Perceptibility Experimental Setup

We use Amazon Mechanical Turk for human testing. Workers were paid to take a survey that first showed an example of an unaltered face and an attacked face with magnitude $\epsilon = 50$. Each worker was then shown 10 images of faces chosen at random, with both unaltered and attacked images. Workers were then asked to choose whether the image had been digitally altered, not digitally altered, or if they could not tell. Each survey was shown to 11 different workers, and the majority label was assigned to the image. This was repeated for 10 different surveys for each attack, resulting in 110 workers surveyed per attack and 100 images voted on per attack (1,100 answers per attack). We selected 100

images randomly from the 500-image LFW subset, where we only kept successful attacks. Human perceptibility was calculated based on the accuracy of each majority label when compared with the true label for each image.

## 5. Results and Discussion

We notice in experiments that there is a correlation between higher attack maximum magnitude $\epsilon$ and human perceptibility. In Figure 2 we can observe this phenomenon in the Square, NES and Bandits curves where perceptibility increases as $\epsilon$ increases. In particular, Square is highly detectable across all tested $\epsilon$ values in part due to the fact that it uses the full magnitude budget for every attack. The NES and Bandits attacks have similar profiles, both being indistinguishably detectable from random guessing (50% detectability) at $\epsilon = 12$, and both slowly increasing and becoming reliably more detectable at $\epsilon = 16$ with around 60% detectability. SimBA exhibits a flatter curve that is more detectable than NES/Bandits at $\epsilon = 12$ and less at $\epsilon = 16$.

In Figure 3, we plot the perceptibility of the attack with varying success rates. In this practical setting, no attack was able to achieve both a 100% success rate and remain reliably imperceptible. The strongest performance was recorded by the Bandits attack, which achieves a high success rate with various epsilon values and the lowest detection rates for most epsilon values. We present statistics for all of the attacks in Table 1. We observe that Bandits obtains the highest success rate at $\epsilon = 20$ of 99.8% (tied with NES and Square) with a human detection rate of 0.54, lower than NES and

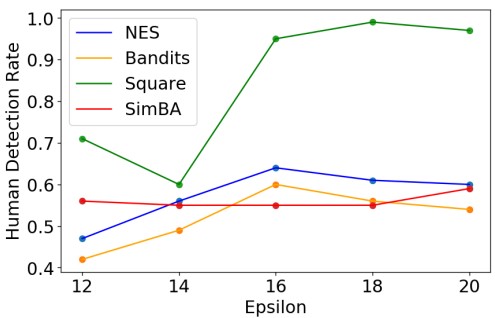

Figure 2. Human attack detection rate across $\epsilon$ values

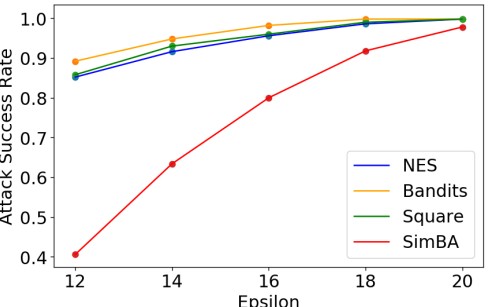

Figure 4. Attack success rate across $\epsilon$ values

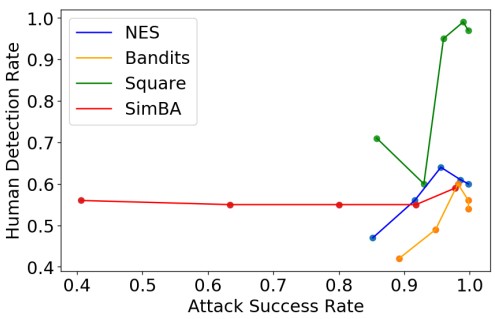

Figure 3. Attack success rate compared with human detection rate

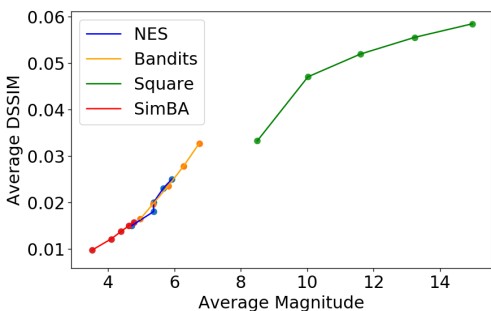

Figure 5. Average magnitude compared with average DSSIM

substantially lower than Square and with an almost comparable number of average queries (1,674 vs. 1,120). We can observe that all of the adapted attacks are successful in attacking images in this new scenario and data distribution, but specifics of success rates, query efficiency, and perceptibility vary compared to the ImageNet setting. For example, even though Square is the latest state-of-the-art attack, Bandits achieves a higher success rate over several $\epsilon$ values. We show examples of the most and least perceptible attacks on faces as annotated by the Mechanical Turk workers in Figure 1. Attacks on images with high skin surface and bright lighting seem to be more perceptible.

In Figure 4, we show success rates across $\epsilon$ values, with most attacks performing similarly in this respect, with slowly increasing curves. The exception is the SimBA attack, which starts with a low success rate and catches up to the others at $\epsilon = 20$. We argue that only looking at this curve, as most works in the adversarial attack space do, does not provide the full picture and it is possible to use human perceptibility tests to more extensively assess attacks.

Finally, in Figure 5, we observe a strong, almost linear correlation between average attack magnitude and average DSSIM. We conclude from this that reporting additional metrics such as DSSIM might be futile, since they are very closely tied to the average attack magnitude, and both do not succeed in capturing human perceptibility.

## 6. Conclusion

Through human perceptibility experiments of black-box attacks on face recognition systems, we have shown in this work that (1) these attacks transfer to a new distribution, but specifics in their success statistics differ, (2) there are trade-offs between attack success rate and human perceptibility, and no attack achieves a 100% success rate while escaping human detection, (3) DSSIM, $\ell_p$ attack magnitude, and other related metrics do not fully capture human perceptibility (4) these metrics are also highly correlated, and reporting several of them for completeness might be unnecessary, and (5) attacks that use the full attack magnitude budget for every attack are much more perceptible in our evaluation. Given these conclusions, we believe it is important for the community to further explore and report the human perceptibility of adversarial attacks and that novel attacks should not be judged solely on traditional metrics.

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
