# OpenReview forum: "Examining the Human Perceptibility of Black-Box Adversarial Attacks on Face Recognition"
_ICML.cc/2021/Workshop/AML — ICML 2021 Workshop AML Poster_

### Official Review · Reviewer_qsNi · 2021-06-20
**Examining the Human Perceptibility of Black-Box Adversarial Attacks on Face Recognition**

**Rating:** Accept
**Confidence:** 5

**Review:**

This paper measures how well black-box attacks adapt to a different data distribution in face verification, and present some insights through human perceptibility experiments of black-box attacks on face recognition systems.  In general, the paper is clearly written, and the author is also encouraged to study the relationship between human perceptibility and attack metrics in more challenging conditions.

---

### Decision · Program_Chairs · 2021-06-21

**Decision:**

Accept (Poster)

**Comment:**

This paper studied the human perceptibility of black-box attacks on face recognition. The paper is well written.